**Data Availability Statement:** The data that support the findings of this study are not publicly available, because the data were obtained from a third party

# Rehospitalization for pneumonia after first pneumonia admission: Incidence and predictors in a population-based cohort study

**Paola Faverio**[1,2]*, **Matteo Monzio Compagnoni**[3,4], **Matteo Della Zoppa**[1,2], **Alberto Pesci**[1,2], **Anna Cantarutti**[3,4], **Luca Merlino**[5], **Fabrizio Luppi**[1,2], **Giovanni Corrao**[3,4]

1 School of Medicine and Surgery, University of Milano-Bicocca, Monza, Italy, 2 Respiratory Unit, San Gerardo Hospital, ASST di Monza, Monza, Italy, 3 National Centre for Healthcare Research & Pharmacoepidemiology, University of Milano-Bicocca, Milan, Italy, 4 Department of Statistics and Quantitative Methods, Laboratory of Healthcare Research & Pharmacoepidemiology, Unit of Biostatistics, Epidemiology and Public Health, University of Milano-Bicocca, Milan, Italy, 5 Regional Health Ministry, Lombardy Region, Milan, Italy

* paola.faverio@unimib.it

## Abstract

### Background and objectives

Hospital readmissions are a frequent complication of pneumonia. Most data regarding readmissions are obtained from the United States, whereas few data are available from the European healthcare utilization (HCU) systems. In a large cohort of Italian patients with a previous hospitalization for pneumonia, our aim was to evaluate the incidence and predictors of early readmissions due to pneumonia.

### Methods

This is a observational retrospective, population based, cohort study. Data were retrieved from the HCU databases of the Italian Lombardy region. 203,768 patients were hospitalized for pneumonia between 2003 and 2012. The outcome was the first rehospitalization for pneumonia. The patients were followed up after the index hospital admission to estimate the hazard ratio, and relative 95% confidence interval, of the outcome associated with the risk factors that we had identified.

### Results

7,275 patients (3.6%) had an early pneumonia readmission. Male gender, age ≥70 years, length of stay of the first admission and a higher burden of comorbidities were significantly associated with the outcome. Chronic use of antidepressants, antiarrhythmics, glucocorticoids and drugs for obstructive airway diseases were also more frequently prescribed in patients requiring rehospitalization. Previous use of inhaled broncodilators, including both beta2-agonists and anticholinergics, but not inhaled steroids, were associated with an increased risk of hospital readmission.

and are available from Lombardy Region. The restriction on data that were used for the current study is imposed by license and agreement between University of Milano Bicocca and Regional Health Authority of Lombardy Region, and so are not publicly available. Data are available upon request from the Lombardy Region. Since data for the present study were shared under an agreement between two parties (special access privileges), requests for information on data access can be directed to Dr. Roberto Blaco, head of the Epidemiologic Observatory of Lombardy Region (contact via roberto_blaco@regione.lombardia.it).

**Funding:** This study was funded by grants from the Italian Ministry of Education, Universities and Research (MIUR) ("PRIN: Progetto di Ricerca di Interesse Nazionale", year 2017, project 2017728JPK). MIUR had no role in the design of the study, the collection, analysis, and interpretation of the data, or the writing of the manuscript.

**Competing interests:** The authors have read the journal's policy and the authors of this manuscript have the following competing interests: G.C. received research support from the European Community (EC), the Italian Medicines Agency (AIFA), and the Italian Ministry of Education, Universities and Research (MIUR). He took part in a variety of projects that were funded by pharmaceutical companies (Novartis, GSK, Roche, AMGEN and BMS). He also received honoraria as a member of the Advisory Board of Roche. No other potential conflicts of interest were declared. This does not alter our adherence to PLOS ONE policies on sharing data and materials.

**Abbreviations:** ATC, anatomical therapeutic chemical codes; HCU, health care utilization; MCS, Multisource Comorbidity Score; NHS, National Health Service; NSAIDs, nonsteroidal anti-inflammatory drugs.

## Conclusions

Frail elderly patients with multiple comorbidities and complex drug regimens were at higher risk of early rehospitalization and, thus, may require closer follow-up and prevention strategies.

## Introduction

Pneumonia is one of the more frequent and potentially serious infectious diseases, being a leading cause of hospitalization worldwide. In the last decades pneumonia hospital admissions have increased by 25–50% in the US and European countries [1,2], particularly in the elderly, who have a 10-fold increased incidence [1,3]. Pneumonia has also been shown to increase short- and long-term mortality [1], mostly affecting frail populations, including elderly and patients with multiple comorbidities [4], substantially heightening the disease burden among adults overall and, thus, becoming one of the main topics in public health.

Age, gender, type of pneumonia, chronic comorbidities and severity of the disease have all been associated with increased short- and long-term mortality in patients with pneumonia [1], especially because pneumonia is a trigger for cardiovascular events and for respiratory exacerbations in predisposed patients [1].

Early hospital readmission (within 30 days) complication after a first hospitalization for pneumonia, with an incidence ranging from 11.8 to 20.8% [5]. Recurring pneumonia, cardiovascular disease and exacerbations of chronic pulmonary diseases are the most common causes of early readmission [6]. Instability factors upon discharge, such as persistent fever and respiratory failure, hospitalization in the previous 90 days and an elevated number of decompensated comorbidities, have all been associated with rehospitalization [7,8]. Nevertheless, also the complexity of medication regimens is a possible predictive factor for readmission in patients affected by pneumonia [9].

Most data concerning readmissions and risk factors come from the United States, because hospitals with higher-than-expected risk adjusted 30-day readmission rates have to deal with major financial penalties [10] imposed by Medicare and Medicaid services since the Hospital Readmission Reduction Program has been implemented in 2012.

With these premises, a large real-world retrospective cohort study was conducted using administrative databases to estimate, in patients who had been previously hospitalized for pneumonia, the incidence of early (within 30 days) pneumonia readmissions and to investigate the predisposing factors for rehospitalizations.

## Materials and methods

### Setting

The present study is based on computerized Healthcare Utilization (HCU) databases of Lombardy, an Italian northern region accounting for almost 10 million people (about 16% of Italy's whole population. In Italy, the National Health Service (NHS) covers the entire resident population, and since 1997 its management in Lombardy has been associated with an automated system of HCU databases that collects and stores a variety of information, including (i) demographic and clinical data on residents who receive NHS assistance, (ii) diagnosis at discharge from public or private hospitals, (iii) reimbursable drug prescriptions dispensed outpatiently or directly administered in hospital, (iv) exemptions from healthcare co-payment for chronic

diseases, and (v) outpatient visits, including specialist visits and diagnostic exams reimbursable by the NHS [11].

A detailed description of the HCU databases of Lombardy for studying the framework of respiratory diseases is available in other previous studies [12–15]. The ICD-9-CM, Anatomical-Therapeutic-Chemical (ATC) and outpatient procedure codes used in the current study are shown in **S1 Table**.

## Cohort selection

Our study was designed according to the procedure shown in **Fig 1**. All 266,766 NHS-eligible residents in Lombardy who had experienced at least one hospital admission with pneumonia as primary or secondary diagnosis (ICD-9-CM code from 480.x to 488.x, 487.x excluded, 4870. x included) during the years 2003 to 2012, were identified. The date of the first hospital discharge was considered as the index date. Four categories of patients were excluded: (i) 50,187 patients aged ≤18 years at the index date; (ii) 2,230 patients who were beneficiaries of the NHS from less than <6 months before the index date; (iii) 438 patients with at least one

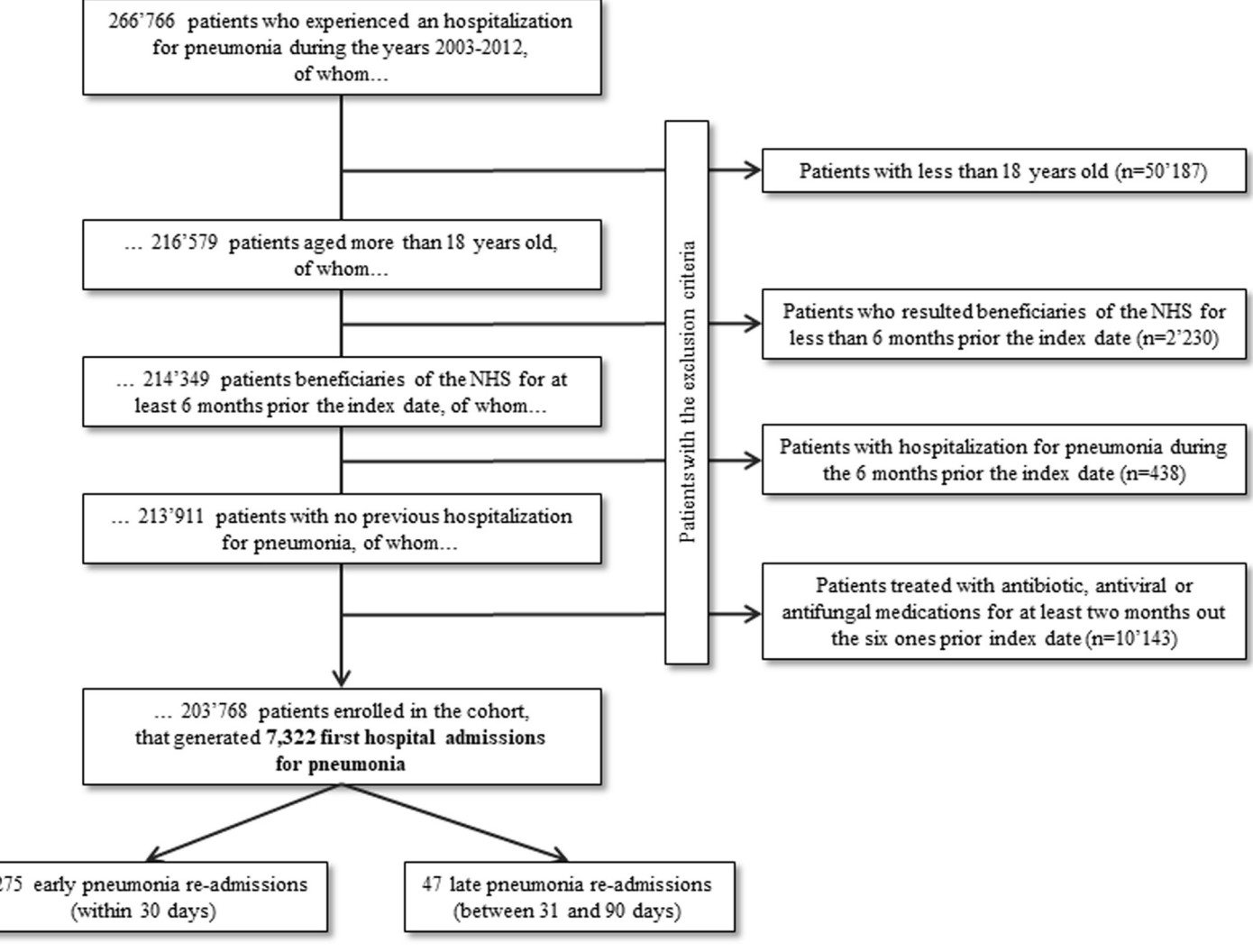

**Fig 1. Flow-chart of inclusion and exclusion criteria for the study cohort selection.** Lombardy region, Italy, 2003–2012. NHS, national health system.

hospitalization for pneumonia in the 6 months before the index date (in order to include only those patients for whom the index admission was not a readmission after previous pneumonia hospitalization); and (iv) 10,143 patients treated cumulatively with antibiotic, antiviral or anti-fungal medications for at least 2 months before the index date. The remaining 203,768 patients were included in the study cohort that accumulated person-years of follow-up from the index date until the earliest date among the occurrence of the outcome (see below) or censoring (occurring at death, emigration, or the 90th day after index date).

## Outcome

The outcome was the first rehospitalization for pneumonia (ICD-9-CM code from 480.x to 488.x, 487.x excluded, 4870.x included) that occurred within 30 days of discharge after the first pneumonia admission; the so-called early readmissions for pneumonia.

## Covariates

The baseline characteristics of cohort members (i.e., those recorded at the index date or during the previous 6 months) included gender, age, length of stay during the index hospitalization, prescriptions for antibiotic, antifungal or antiviral drugs at discharge, previous admissions for lung cancer or chronic lung disease and previous prescriptions for other drugs (antidepressants, antiarrhythmics, antithrombotics, antihypertensives, inhaled steroid drugs and bronchodilators, nonsteroidal anti-inflammatory drugs (NSAIDs), digoxin, statins and benzodiazepines). Some of these latter medications can be considered as proxies for underlying chronic medical conditions.

Other clinical and therapeutic characteristics were also measured during the index hospital stay: venous thromboembolism, thrombocytosis, *Clostridium difficile* infection, pressure ulcer, lung procedure, respiratory failure, mechanical ventilation use, ≥1 unit of blood transfusion.

Finally, we used the Multisource Comorbidity Score (MCS), a new comorbidity index obtained from inpatient diagnostic information and outpatient drug prescriptions, validated using data from Lombardy and other regions of Italy [11]. Patients were categorized as having low (0–9) or high (≥10) MCS score.

## Data analysis

Demographic data, clinical characteristics and therapeutic regimens were compared among patients who had a pneumonia readmission and those who did not. A cohort design was implemented and a Cox proportional hazard model was fitted to estimate hazard ratios (HR), and relative 95% confidence intervals (CI), for the association between the selected potential predisposing factors and the risk of early readmission for pneumonia. The estimates were adjusted by those covariates listed in the "Covariates" section. The effect of the predisposing factors on the risk of a pneumonia rehospitalization was evaluated both for the entire cohort and, separately, for age classes (<70 and ≥70 years).

Finally, the robustness of estimates regarding potential bias introduced by unmeasured confounders (e.g., smoking status) was investigated by using the rule-out approach described by Schneeweiss [16]. This approach involves detecting the amount of the overall confounding required to fully account for the exposure–outcome association, thus moving the observed point estimate to null.

The Statistical Analysis System software (version 9.4; SAS Institute, Cary, North Carolina, USA) was used to perform the analyses. For all hypotheses tested, two-tailed p-values <0.05 were considered to be significant.

### Ethical approval and consent to participate

According to the rules from the Italian Medicines Agency (available at: http://www.agenziafarmaco.gov.it/sites/default/files/det_20marzo2008.pdf), retrospective studies using administrative databases do not require Ethics Committee protocol approval. Furthermore, according to General Authorization for the Processing of Personal Data for Scientific Research Purposes issued by the Italian Privacy Authority on August 10, 2018 (available at: https://www.garanteprivacy.it/web/guest/home/docweb/-/docweb-display/docweb/9124510) this study was exempt from informed consent. In order to protect privacy, and to guarantee individual records anonymity, after the record-linkage between HCU databases and the data extraction procedure, the individual identification codes were automatically converted into anonymous by the regional IT technicians, so that researchers had access to full anonymized data.

## Results

### Study population

The 203,768 cohort members accumulated 44,226 person-years of follow-up, on average 0.22 years per patient. During this period 33,746 patients (16.6%) experienced a 30-day all-cause hospital readmission, whereas 7,275 patients (3.6%) required an early pneumonia readmission (with an incidence rate of 16.4 cases every 100 person-years).

Demographic data, clinical characteristics and therapeutic regimens of patients undergoing a first hospitalization for pneumonia between the years 2003 and 2012 in the whole cohort, and according to whether or not patients experienced the outcome of interest, are summarised in Table 1. Patients who required rehospitalization were more frequently male, with a longer length of hospital stay during the first pneumonia admission and a higher burden of comorbidities, measured by the MCS (**Table 1**).

Chronic renal and lung diseases, as well as previous cardiovascular events and history of cancer, were more frequent in patients rehospitalized for pneumonia compared to those who were not. Chronic use of medications—antidepressants, antiarrhythmics, inhaled drugs for obstructive airway diseases, including steroids and bronchodilators, and persistent systemic glucocorticoid therapy—were more frequently prescribed in patients requiring rehospitalization (**Table 1**).

Factors complicating the first hospitalization, such as respiratory failure and mechanical ventilation use, pressure ulcers and need of blood transfusions, were more frequent among patients requiring a readmission.

### Risk factors for pneumonia rehospitalization

Possible risk factors for early pneumonia rehospitalizations are summarised in **Table 2**. Male gender, age ≥70 years, longer length of hospital stay during the first hospitalization and a MCS ≥10 were all associated with an increased risk of pneumonia rehospitalization. Previous medications—inhaled bronchodilators, including both beta2-agonists and anticholinergics—were associated with increased risk of hospital readmission, whereas this association was not observed for inhaled steroids, either alone or in association with broncodilators. Chronic aspirin and systemic glucocorticoid therapy were also risk factors for pneumonia readmissions, but NSAID therapy was not. As for medical history, renal and chronic lung disease were associated with an increased risk of pneumonia rehospitalization.

### Risk factors for pneumonia rehospitalization, stratified by age

When stratifying the analysis by age classes (≥70 years vs. <70 years), similar risk factors were observed, with the exception of complications during hospital stay (**Fig 2**). C. difficile infection

**Table 1. Baseline demographic and clinical characteristics and therapeutic regimens of the 203,768 patients considered in the study, in the whole cohort and according to whether they experienced the outcome of interest.** Lombardy, Italy, 2003–2012.

| | Whole cohort (n = 203,768) | Pneumonia rehospitalization (n = 7,275) | No pneumonia rehospitalization (n = 196,493) | P value |
|---|---|---|---|---|
| **Demographics** | | | | |
| Male (%) | 112,698 (55.3) | 4,509 (62.0) | 108,189 (55.1) | < .0001 |
| Age, years: mean (SD) | 71.2 (16.3) | 71.6 (16.2) | 71.2 (16.3) | 0.3486 |
| Length of stay (LOS): mean (SD) | 15.4 (16.2) | 16.5 (16.9) | 15.4 (16.1) | < .0001 |
| Multisource Comorbidity Score [α] (%) | | | | |
| Low | 139,732 (68.6) | 4,579 (62.9) | 135,153 (68.8) | < .0001 |
| High | 64,036 (31.4) | 2,696 (37.1) | 61,340 (31.2) | |
| **Previous diseases (%)** | | | | |
| CV disease | 115,118 (56.5) | 4,305 (59.2) | 110,813 (56.4) | < .0001 |
| Cerebrovascular disease | 27,034 (13.3) | 1,015 (14.0) | 26,019 (13.2) | 0.0795 |
| Liver disease | 9,154 (4.5) | 343 (4.7) | 8,811 (4.5) | 0.3510 |
| Renal disease | 14,765 (7.3) | 680 (9.4) | 14,085 (7.2) | < .0001 |
| Cancer | 34,263 (16.8) | 1,543 (21.2) | 32,720 (16.7) | < .0001 |
| Lung cancer | 4,170 (12.2) | 175 (2.4) | 3,995 (2.0) | 0.0276 |
| Chronic lung disease | 31,369 (15.4) | 1,205 (16.6) | 30,164 (15.4) | 0.0049 |
| **Previous medications (%)** | | | | |
| Drugs for obstructive airway diseases | 46,239 (22.7) | 1,781 (24.5) | 44,458 (22.6) | 0.0002 |
| Inhaled steroids | 35,584 (17.5) | 1,388 (19.1) | 34,196 (17.4) | 0.0002 |
| Inhaled broncodilators [β] | 29,012 (14.2) | 1,186 (16.3) | 27,826 (14.16) | < .0001 |
| Persistent aspirin therapy | 1,972 (1.0) | 91 (1.3) | 1,881 (1.0) | 0.0120 |
| Persistent NSAID therapy [γ] | 4,889 (2.4) | 178 (2.5) | 4,711 (2.4) | 0.7877 |
| Persistent glucocorticoid therapy | 4,037 (2.0) | 184 (2.5) | 3,853 (2.0) | 0.0006 |
| Antidepressants | 26,374 (12.9) | 1,021 (14.0) | 25,353 (12.9) | 0.0047 |
| Antiarrhythmics | 15,028 (7.4) | 647 (8.9) | 14,381 (7. 3) | < .0001 |
| Statins | 28,751 (14.1) | 999 (13.7) | 27,752 (14.1) | 0.3460 |
| Antihypertensives | 120,543 (59.2) | 4,348 (59.8) | 116,195 (59.1) | 0.2816 |
| **During hospital stay (%)** | | | | |
| Venous thromboembolism | 882 (0.4) | 32 (0.4) | 850 (0.4) | 0.9260 |
| *Clostridium difficile* infection | 2,031 (1.0) | 93 (1.3) | 1,938 (1.0) | 0.0138 |
| Pressure ulcer | 2,371 (1.2) | 115 (1.6) | 2,256 (1.2) | 0.0007 |
| Chest / lung surgery § | 17,190 (8.4) | 772 (10.6) | 16,418 (8.4) | < .0001 |
| Respiratory failure | 33,940 (16.7) | 1,604 (22.1) | 32,336 (16.5) | < .0001 |
| Mechanical ventilation | 11,879 (5.8) | 698 (9.6) | 11,181 (5.7) | < .0001 |
| ≥ 1 units blood transfusion | 13,716 (6.7) | 707 (9.7) | 13,009 (6.6) | < .0001 |

Abbreviation: SD, standard deviation; CV, cardiovascular; NSAID, nonsteroidal anti-inflammatory drug

[α] The clinical status was assessed by the Multisource Comorbidity Score (MCS) according to the hospital admission and the drugs prescribed in the six-months period before the index date. Two categories of clinical status (MCS) were considered: good (Low MCS: score < 10) and poor (High MCS: score ≥ 10).

[β] Inhaled broncodilators: both beta2-agonists and anticholinergics.

§ Chest / lung surgery: all operations on the respiratory system, including excision of lung and bronchus, but excluding chest drainage and bronchoscopy.

was associated with increased risk of readmission for pneumonia only in elderly patients. A need for blood transfusion appeared to be a stronger risk factor in younger patients.

**Table 2. Hazard Ratios (HR), and relative 95% Confidence Intervals (CI), of early rehospitalizations for pneumonia in the whole cohort (203,768 patients), estimated by a multivariable Cox proportional hazard model.**

| Covariates | HR (95% CI) |
|---|:---:|
| **Baseline** | |
| Sex | |
| Female | 1.00 |
| Male | 1.31 (1.25–1.37) |
| Age (years) | |
| <70 | 1.00 |
| ≥70 | 1.11 (1.05–1.17) |
| Length of stay (days) | |
| <10 | 1.00 |
| ≥10 | 1.10 (1.05–1.17) |
| Multisource Comorbidity Score [α] | |
| Low | 1.00 |
| High | 1.23 (1.16–1.30) |
| **During hospital stay** | |
| Respiratory failure | 1.35 (1.26–1.43) |
| Mechanical ventilation | 1.50 (1.37–1.64) |
| ≥1 unit blood transfusion | 1.44 (1.33–1.56) |
| Pressure ulcer | 1.34 (1.11–1.61) |
| **Previous medications** | |
| Drugs for obstructive airway diseases | 0.90 (0.79–1.03) |
| Inhaled steroids | 1.08 (0.96–1.21) |
| Inhaled broncodilators | 1.14 (1.03–1.26) |
| Persistent aspirin therapy | 1.30 (1.05–1.60) |
| Persistent NSAID therapy | 1.03 (0.88–1.19) |
| Persistent glucocorticoid therapy | 1.21 (1.04–1.40) |
| Antidepressants | 1.09 (1.01–1.16) |
| Antiarrhythmics | 1.14 (1.05–1.24) |
| **Previous diseases** | |
| Liver disease | 0.99 (0.89–1.11) |
| Renal disease | 1.19 (1.09–1.29) |
| Lung cancer | 0.98 (0.84–1.15) |
| Chronic lung disease | 1.12 (1.03–1.25) |

[α] The clinical status was assessed by the Multisource Comorbidity Score (MCS) according to the hospital admission and the drugs prescribed in the six-months period before the index date. Two categories of clinical status (MCS) were considered: good (Low MCS: score < 10) and poor (High MCS: score ≥ 10).

## Sensitivity analysis

The robustness of estimates regarding bias introduced by an unmeasured confounder, like smoking status, was evaluated using the rule-out approach (**Fig 3**). We performed this sensitivity analysis taking into account two of those predisposing factors that were found to have a significant association with the outcome, as well as being associated with the unmeasured confounder (i.e., respiratory failure and mechanical ventilation use during the index hospitalization). The rule-out approach was applied to evaluate if the observed harmful effect of those two risk factors was overinflated by the unmeasured confounder. For example, patients with a respiratory failure during the index hospitalization had a 3-fold smokers' prevalence than those who

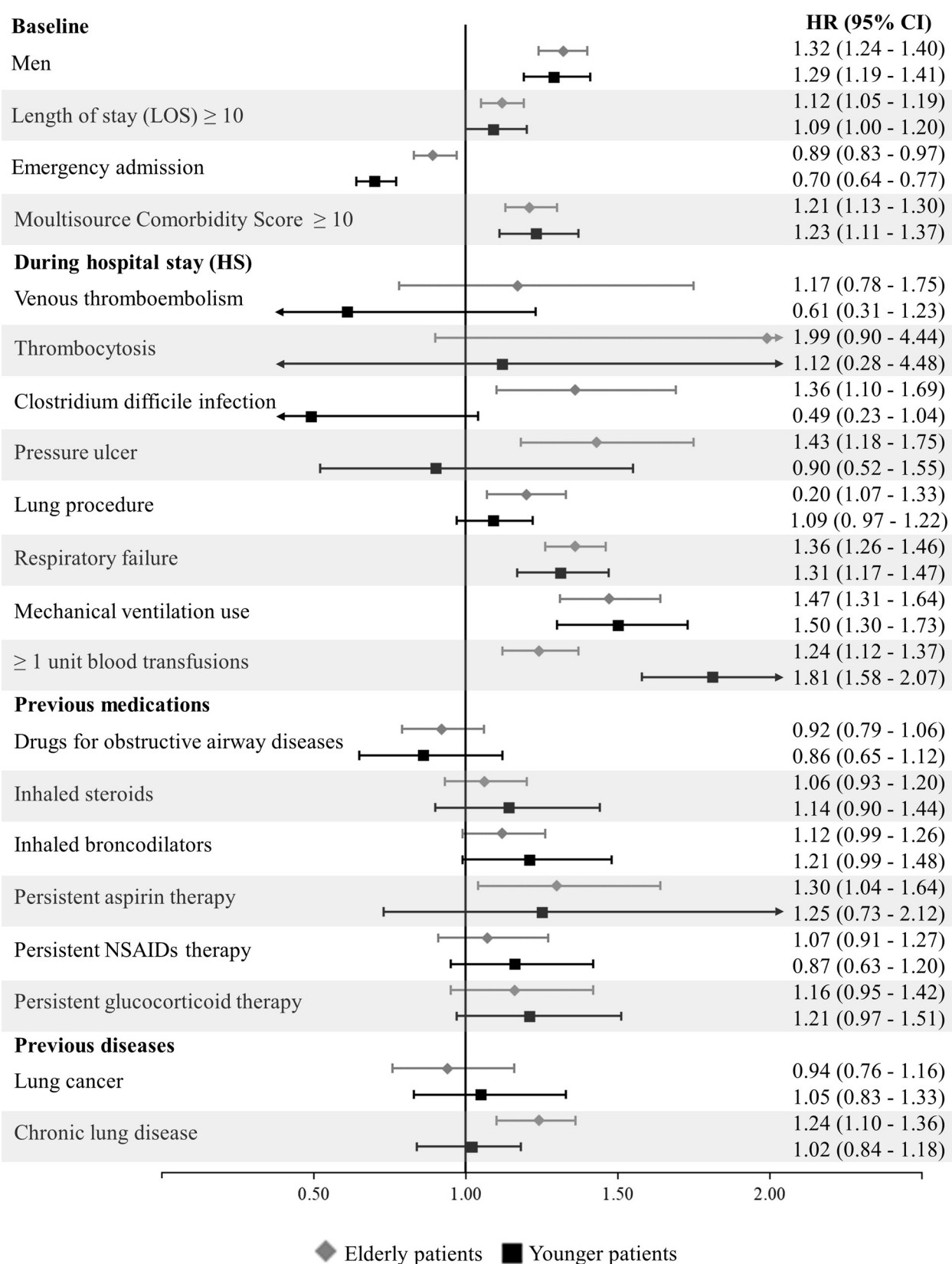

| | HR (95% CI) |
|---|---|
| **Baseline** | |
| Men | 1.32 (1.24 - 1.40) |
| | 1.29 (1.19 - 1.41) |
| Length of stay (LOS) ≥ 10 | 1.12 (1.05 - 1.19) |
| | 1.09 (1.00 - 1.20) |
| Emergency admission | 0.89 (0.83 - 0.97) |
| | 0.70 (0.64 - 0.77) |
| Moultisource Comorbidity Score ≥ 10 | 1.21 (1.13 - 1.30) |
| | 1.23 (1.11 - 1.37) |
| **During hospital stay (HS)** | |
| Venous thromboembolism | 1.17 (0.78 - 1.75) |
| | 0.61 (0.31 - 1.23) |
| Thrombocytosis | 1.99 (0.90 - 4.44) |
| | 1.12 (0.28 - 4.48) |
| Clostridium difficile infection | 1.36 (1.10 - 1.69) |
| | 0.49 (0.23 - 1.04) |
| Pressure ulcer | 1.43 (1.18 - 1.75) |
| | 0.90 (0.52 - 1.55) |
| Lung procedure | 0.20 (1.07 - 1.33) |
| | 1.09 (0.97 - 1.22) |
| Respiratory failure | 1.36 (1.26 - 1.46) |
| | 1.31 (1.17 - 1.47) |
| Mechanical ventilation use | 1.47 (1.31 - 1.64) |
| | 1.50 (1.30 - 1.73) |
| ≥ 1 unit blood transfusions | 1.24 (1.12 - 1.37) |
| | 1.81 (1.58 - 2.07) |
| **Previous medications** | |
| Drugs for obstructive airway diseases | 0.92 (0.79 - 1.06) |
| | 0.86 (0.65 - 1.12) |
| Inhaled steroids | 1.06 (0.93 - 1.20) |
| | 1.14 (0.90 - 1.44) |
| Inhaled broncodilators | 1.12 (0.99 - 1.26) |
| | 1.21 (0.99 - 1.48) |
| Persistent aspirin therapy | 1.30 (1.04 - 1.64) |
| | 1.25 (0.73 - 2.12) |
| Persistent NSAIDs therapy | 1.07 (0.91 - 1.27) |
| | 0.87 (0.63 - 1.20) |
| Persistent glucocorticoid therapy | 1.16 (0.95 - 1.42) |
| | 1.21 (0.97 - 1.51) |
| **Previous diseases** | |
| Lung cancer | 0.94 (0.76 - 1.16) |
| | 1.05 (0.83 - 1.33) |
| Chronic lung disease | 1.24 (1.10 - 1.36) |
| | 1.02 (0.84 - 1.18) |

◆ Elderly patients   ■ Younger patients

**Fig 2. Age-stratified Hazard ratios (HR), and relative 95% confidence intervals (CI), of early rehospitalizations for pneumonia associated with selected risk factors, estimated with a multivariable Cox proportional hazard model.** Lombardy region, Italy, 2003–2012. Estimates, stratified for age classes (<70 and ≥70 years), were obtained through a multivariable Cox proportional hazard model. For each covariate of which HR of the outcome was reported in the figure, the reference category was that of unexposed patients to this covariate (e.g., for persistent aspirin therapy the reference category was that of patients who was not exposed to a persistent aspirin therapy). NSAIDs, nonsteroidal anti-inflammatory drugs.

had not a respiratory failure (exposure-confounder odds ratio, $OR_{EC}$, = 3). In order to nullify the observed harmful effect of the respiratory failure, smoking status should increase the risk of experiencing an hospital readmission for pneumonia by 3-fold (confounder-outcome relative

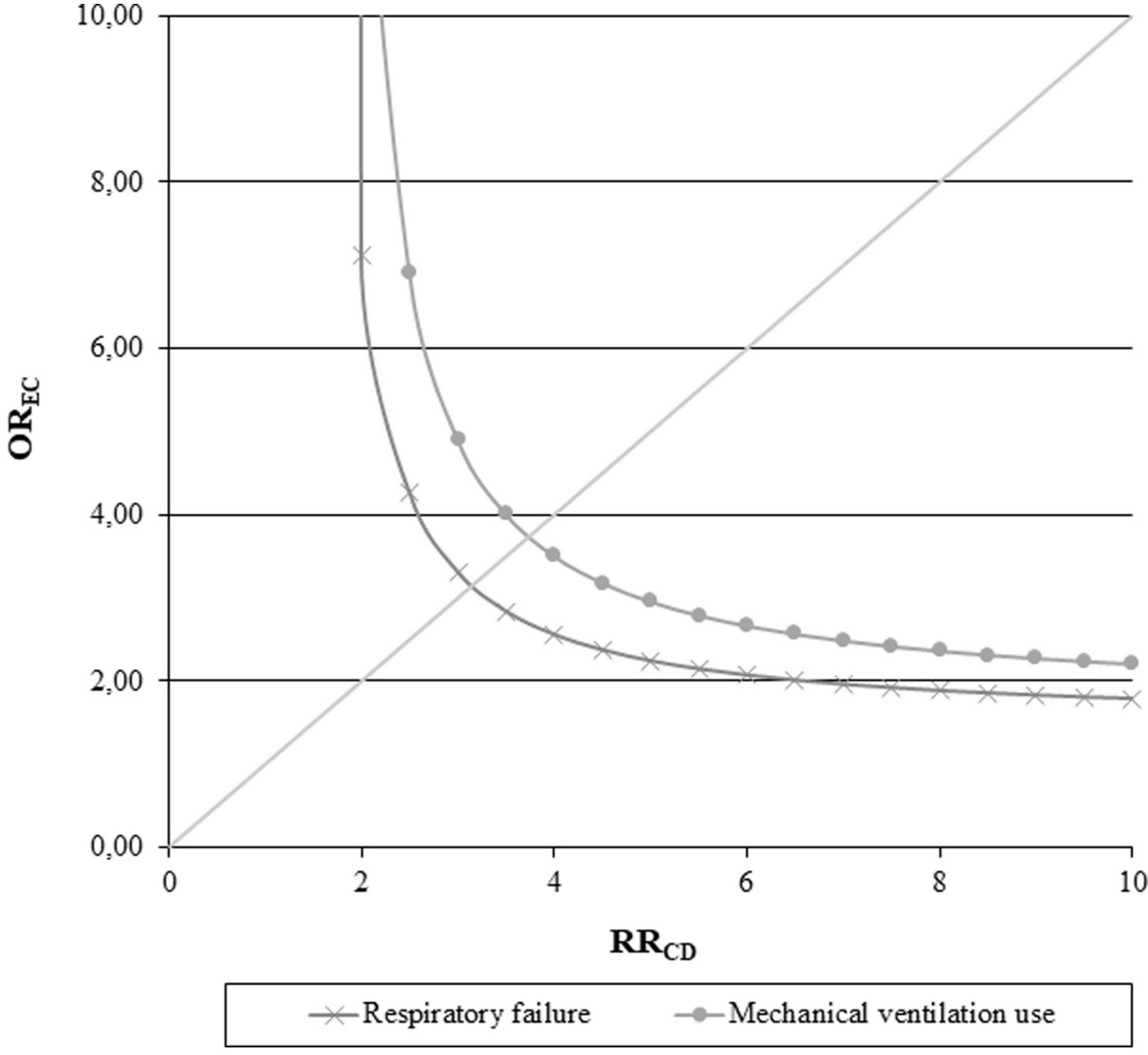

**Fig 3. Influence of smoking status as an unmeasured confounder on the association between respiratory failure (or mechanical ventilation use) during the index hospital stay and the risk of experiencing a pneumonia readmission.** Lombardy region, Italy, 2003–2012. The graph indicates the combinations of confounder-outcome and exposure-confounder associations that would be required to move the observed effect of the considered risk factors towards the null. We set the possible generic unmeasured confounder, e.g., smoking status: (i) to have a 21% prevalence of exposure among patients hospitalized for pneumonia [17], (ii) to increase the risk of the outcome onset up to 10-fold more in patients exposed to the confounder than in those not exposed, and (iii) to be up to 10-fold more common among patients exposed to the predisposing factor than in those not exposed.

risk, $RR_{CD}$, = 3). On the other hand, admitting that smoking increases the risk of the outcome by 2-fold, prevalence of smokers among patients who experienced a respiratory failure should be 6-fold higher than those who did not, in order to nullify the observed effect.

Similar results and interpretation, if not even more marked and less likely, can also be observed considering the mechanical ventilation use.

## Discussion

In the cohort of 203,768 adult patients hospitalized for pneumonia in the northern Italian region of Lombardy between 2003 and 2012, the majority were elderly, with a high burden of comorbidities (particularly cardiovascular events) and on treatment with multiple medications.

Out of these 203,768 patients, 33,746 (16.6%) experienced an all-cause rehospitalization within 30 days after the index hospital discharge, meaning that one in six patients hospitalized for pneumonia was readmitted for any cause within 30 days of discharge, according to the results reported in several observational studies. In literature, all-cause 30-day readmission rates after a first pneumonia hospitalization ranged from 8.6 to 20.8% [8,18], whereas many studies that have focused on the Medicare population showed all-cause 30-day readmission rate of 17%-25% [19,20].

Furthermore, in the present study pneumonia-related hospital readmissions accounted for about 22% of total 30-day readmissions, e.g., 7,275 patients (3.6%) required an early readmission (within 30 days of hospital discharge) for a new case of pneumonia.

In a multivariate analysis, the main risk factors for early rehospitalization were male gender, older age, higher burden of comorbidities and longer length of hospital stay during the first pneumonia hospitalization.

The most interesting finding of the present study was the overall readmission rate occurring in the first 30 days (3.6%), leading to the conclusion that patients in the first month are at risk for infectious complications and therefore may require a close follow-up. Most recent pneumonia guidelines do not suggest a follow-up of patients discharged from the hospital, with the exception of the British Thoracic Society guidelines, which suggest arranging a clinical review by the general practitioner or in a hospital clinic at ~6 weeks for all patients [21]. According to our results, a timely follow-up within 4 weeks of hospital discharge especially among patients who can most benefit from this intervention (i.e., elderly patients with a high burden of comorbidities including chronic lung diseases) is suggested. The incidence of 30-day pneumonia readmissions differs between European countries and increases with age in all countries. In most cases, pneumonia affects already frail populations, including the elderly and those with multiple chronic conditions. Thus, rehospitalizations, including those for pneumonia, load an additional burden on these vulnerable population. Therefore, various interventions aimed at decreasing the risk of hospital readmissions by targeting transitional and territorial care and post-discharge care coordination are needed.

In our study, inhaled broncodilators were associated with an increased risk of hospital readmission, whereas inhaled steroids, either alone or in association with bronchodilators, were not associated to the outcome; however, only 15% of patients in our cohort had chronic lung diseases. In contrast, the association between the use of inhaled steroids and the risk of developing pneumonia has been described in patients with chronic obstructive pulmonary disease, particularly those receiving fluticasone, but it is still a matter of debate [22–24].

Although our results seem to support the safe use of inhaled steroids in association to the risk of rehospitalization for pneumonia in the general population, they may not apply to the subgroup of patients with specific clinical conditions, such as chronic lung disease.

Chronic aspirin therapy were also found to be risk factors for pneumonia readmissions, whereas this association was not observed for chronic NSAID therapy. Aspirin has been considered a possible add-on therapy in patients with pneumonia and significant coronary risk factors because of its preventive role in cardiovascular events [25]. These are common complications in patients with community-acquired pneumonia, particularly those with hypoxemia and high systemic inflammatory response [25]. However, NSAID therapy, including the use of aspirin, prior hospital referral for pneumonia has been associated with an increased number of pleuropulmonary complications, such as pleural empyema and lung cavitations [26,27]. The association found in our population between chronic aspirin use and pneumonia readmissions may be due to those delayed pleuropulmonary complications.

Chronic use of systemic steroids, as well as other immunosuppressive therapies not included in this study because of the small number of cases, is also a recognized risk factor for pneumonia [28].

Despite the high population of elderly patients in our study, we found no difference in risk factors for rehospitalization between younger and older patients, with the exception of *C. difficile* infection and a need for blood transfusions during the index hospital stay. In the elderly, an acute disease such as pneumonia can cause a loss of physiologic reserve [29] that may appear as various complications, such as *C. difficile* infection, in those who require prolonged antibiotic therapies [30].

Our study has several elements of strength: (i) the investigation was based on a very large unselected population, which was made possible because of the cost-free health care system in Italy, which covers virtually all citizens; (ii) the drug prescription database provided highly accurate data, because pharmacists are required to report prescriptions in detail to obtain reimbursement, and incorrect reports about the dispensed drugs have legal consequences [31]; (iii) our study is one of the firsts to describe the incidence and risk factors for rehospitalization due to pneumonia in a population of patients with a first hospitalization for pneumonia; (iv) we excluded patients who had been treated with antibiotic, antiviral or antifungal medications for at least 2 months before the first pneumonia hospitalization. By doing so, we ruled out patients with chronic infections that might have led to an overestimation of the outcome. (v) We excluded patients who experienced a hospitalization for pneumonia in the 6 months before the index date; with this cautionary criterium we ensured the inclusion in the cohort of patients for whom the index admission was not a rehospitalization after previous pneumonia hospital admission, avoiding a potential misclassification of the outcome. Finally, (vi) a sensitivity analysis for the presence of unmeasured confounding confirmed the robustness of findings provided by the main analysis.

Our investigation also has several limitations beyond those inherent the observational studies. A main limitation is that, because of privacy regulations, hospital records were not available for scrutiny, which means that the diagnostic validity of pneumonia could not be checked. Another limitation of our study is that, although the patients' clinical status can be inferred (and the data adjusted for) from knowledge of hospitalizations, treatments for pneumonia and pulmonary diseases and assumption of antibiotic, antifungal, antiviral therapy and other drugs, information does not include blood pressure, fever, respiratory rate, new-onset confusion, severity of pneumonia, and other clinical variables. Thus, as with any observational investigation, our findings may be affected by unmeasured confounding factors. However, the sensitivity analysis we carried out suggest that it is unlikely that the results of the study could be confounded by factors not measurable in the HCU databases (e.g., smoking status).

Furthermore, the data analyzed in this study relate to several years ago and are not updated, for this reason our study may not have been able to include some changes that have recently occurred in the care of hospitalized patient for pneumonia. However, our all-case 30-day

readmission rates after a first pneumonia hospitalization agree with those reported in recent observational studies, ranging from 8.6 to 25% [6,8,19,20], thus we can likely suppose that all-case 30-day readmission rates after a first pneumonia hospitalization remained stable.

These limitations notwithstanding, our investigation offers quantitative evidence that patients in the first 30 days after discharge are at higher risk of hospital readmission for pneumonia. Frail patients (especially the elderly, who have a higher burden of comorbidities and chronic use of medications) are at higher risk of rehospitalization and, thus, may require closer follow-up and prevention strategies.

Future perspectives should include a better characterisation of chronic medication use in correlation with the underlying disease to evaluate prescriptive appropriateness. A better knowledge of risk factors for pneumonia rehospitalization should guide large-scale prevention efforts.

## Supporting information

**S1 Table. ICD-9-CM diagnostic, Anatomical-Therapeutic-Chemical (ATC) medication and outpatient procedure codes considered in the current study.** Lombardy region, Italy, 2003–2012.
(DOC)

## Author Contributions

**Conceptualization:** Paola Faverio, Matteo Monzio Compagnoni, Alberto Pesci, Anna Cantarutti, Giovanni Corrao.

**Data curation:** Paola Faverio, Matteo Monzio Compagnoni, Matteo Della Zoppa, Alberto Pesci, Luca Merlino, Giovanni Corrao.

**Formal analysis:** Paola Faverio, Matteo Monzio Compagnoni, Anna Cantarutti, Luca Merlino, Giovanni Corrao.

**Funding acquisition:** Giovanni Corrao.

**Investigation:** Paola Faverio, Matteo Monzio Compagnoni, Matteo Della Zoppa, Anna Cantarutti, Luca Merlino, Giovanni Corrao.

**Methodology:** Paola Faverio, Matteo Monzio Compagnoni, Luca Merlino, Giovanni Corrao.

**Project administration:** Paola Faverio.

**Resources:** Paola Faverio, Luca Merlino, Giovanni Corrao.

**Software:** Anna Cantarutti, Luca Merlino, Giovanni Corrao.

**Supervision:** Giovanni Corrao.

**Writing – original draft:** Paola Faverio, Matteo Monzio Compagnoni, Matteo Della Zoppa, Alberto Pesci, Anna Cantarutti, Fabrizio Luppi.

**Writing – review & editing:** Paola Faverio, Matteo Monzio Compagnoni, Matteo Della Zoppa, Alberto Pesci, Anna Cantarutti, Luca Merlino, Fabrizio Luppi, Giovanni Corrao.

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
