## [Decision Letter · Decision Letter 0]

17 Feb 2020

PONE-D-19-32710

Rehospitalization for pneumonia after first pneumonia admission: Incidence and predictors in a population-based cohort study

PLOS ONE

Dear Dr. Faverio,

Thank you for submitting your manuscript to PLOS ONE. After careful consideration, we feel that it has merit but does not fully meet PLOS ONE’s publication criteria as it currently stands. Therefore, we invite you to submit a revised version of the manuscript that addresses the points raised during the review process.

Thank you for your submission to PLOS ONE. You will note that both reviewers identified a number of strengths as well as a number of limitations to your paper. In most cases, the noted limitations relate to some inherent issue in the administrative data or coding; and while it is recognized that you will not be able to address these in your analysis, some further nuanced discussion of these issues is warranted. Further, you will see that both reviewers picked up the 90-days readmissions - where one reviewer thought further discussion was needed and the other thought that the 90-day readmission analysis should be dropped altogether. I will leave it to you and your co-authors to decide; however, if exploring 90-day readmissions had been part of your original objectives then I would encourage to consider keeping it but better explain why no additional analyses are presented.

We would appreciate receiving your revised manuscript by Apr 02 2020 11:59PM. To enhance the reproducibility of your results, we recommend that if applicable you deposit your laboratory protocols in protocols.io, where a protocol can be assigned its own identifier (DOI) such that it can be cited independently in the future. For instructions see: http://journals.plos.org/plosone/s/submission-guidelines#loc-laboratory-protocols

We look forward to receiving your revised manuscript.

Kind regards,

Andrea Gruneir

Academic Editor

PLOS ONE

Journal Requirements:

2. We noticed you have some minor occurrence(s) of overlapping text with the following previous publication(s), which needs to be addressed:

https://doi.org/10.1016/j.ejim.2017.09.023

https://doi.org/10.1002/pds.4206

https://doi.org/10.1016/j.ejim.2017.09.023

https://doi.org/10.1186/s12883-017-0796-3

In your revision ensure you cite all your sources (including your own works), and quote or rephrase any duplicated text outside the Methods section. Further consideration is dependent on these concerns being addressed.

3. In the ethics statement in the manuscript and in the online submission form, please provide additional information about the patient records/samples used in your retrospective study. Specifically, please ensure that you have discussed whether all data/samples were fully anonymized before you accessed them and/or whether the IRB or ethics committee waived the requirement for informed consent. If patients provided informed written consent to have data/samples from their medical records used in research, please include this information.

"I have read the journal's policy and the authors of this manuscript have the following competing interests: G.C. received research support from the European Community (EC), the Italian Medicines Agency (AIFA), and the Italian Ministry of Education, Universities and Research (MIUR). He took part in a variety of projects that were funded by pharmaceutical companies (Novartis, GSK, Roche, AMGEN and BMS). He also received honoraria as a member of the Advisory Board of Roche. No other potential conflicts of interest were declared."

6. Your ethics statement must appear in the Methods section of your manuscript. If your ethics statement is written in any section besides the Methods, please move it to the Methods section and delete it from any other section. Please also ensure that your ethics statement is included in your manuscript, as the ethics section of your online submission will not be published alongside your manuscript.

Reviewers' comments:

Reviewer's Responses to Questions

**Comments to the Author**

1. Is the manuscript technically sound, and do the data support the conclusions?

Reviewer #1: Partly

Reviewer #2: Yes

2. Has the statistical analysis been performed appropriately and rigorously? 

Reviewer #1: Yes

Reviewer #2: I Don't Know

3. Have the authors made all data underlying the findings in their manuscript fully available?

Reviewer #1: Yes

Reviewer #2: Yes

4. Is the manuscript presented in an intelligible fashion and written in standard English?

Reviewer #1: Yes

Reviewer #2: Yes

5. Review Comments to the Author

Reviewer #1: This study shows the factors associated with readmissions in a large cohort of patients with pneumonia. Also, this study adds evidence about complexity medication regimens as a possible predictive factor for readmission.

The follow-up of patients included early and late readmission (30 and 90 days). Unfortunately, the analysis of patients who had a late readmission were not presented. According to your study aimed, information should be provided regarding patient characteristics and associated factors with late readmissions.

The study data were retrieved from the health care utilization database of NHS. Community-acquired pneumonia and hospital-acquired pneumonia were included? Please, add the pneumonia criteria and ICD-9CM codes includes of index admission. Interestingly, perhaps you have include information about type of pneumonia and causative microorganism.

You need explain if a collinearity analysis was perform. Why did you not show the use of statins (Table 1)?

Tables, figures a references need a review (references duplicates nº 5 and 14)

The introduction and discussion needs a little information regarding frail early patients.

Reviewer #2: The authors present a retrospective review of a large cohort of patients admitted with pneumonia to multiple hospitals in Northern Italy between 2003 and 2012 in which they assess the rate of pneumonia-specific readmission and its contributing cofactors. As opposed to most other studies on pneumonia readmission rates, the authors focus on related, pneumonia-specific readmissions, rather than all-cause readmissions. This is a strength of the study. Limitations include the significant delay between data collection and manuscript.

Major critiques

The authors should explain why they feel it is relevant to consider readmissions that occur beyond 30 days in their analysis. Intuitively it seems that these so called “late readmissions” are unrelated to factors involved in the original case. Furthermore the rate of late readmissions is very low. Unless the authors can add further justification for including this in the paper I would suggested removal and focusing on the readmissions within 30 days which is the standard most other research in this area examines.

The data being analyzed in this study is currently 8 – 17 years old. Certainly temporal changes in the care of hospitalized patients have transpired, and the fact that the conclusions are based on old data should be clarified as a limitation in the discussion sections. Ideally the authors would frame their results against comparative readmission rates from more recent data and put their results into context. Unfortunately few studies exist with pneumonia specific readmission rates, but if the authors have access to all-cause 30 day readmission rates, they could make the case for whether temporal trends suggest increasing, decreasing or stable rates compared to the study period.

In Table 1, there appears to be significant differences in various comorbidities between readmitted and not readmitted patients; this is pointed out in line 15 on p 9. However when adjusted hazard ratios are calculated (Table 2) only renal disease appears to be a predictor. In the Results section (p10, line 4-7) the authors note that use of meds for chronic lung disease were associated with increased readmission risk, but chronic lung disease as a comorbidity was not significant. Aren't some of these meds in fact "proxies" for the underlying chronic medical conditions? The authors should discuss these dichotomies and the implications for the limitations of their data in the discussion section.

The paper would benefit from more careful proofreading and correction of grammatical errors, and some effort to improve the clarity of writing.

Data Table 1 lacks column headings, these should be added.

Minor critiques

Late readmissions are mentioned in the introduction but given the lack of clear clinical relevance and the exceedingly small number identified by the author, I think the paper would be more effective if this detail was omitted throughout the manuscript.

The statement about readmission penalties in the US is inaccurate in that it refers to the “private health insurance system.” In fact, 30-day readmission penalties in the US are imposed by Medicare which is a federally funded health insurance program for adults age 65 and older. It is not private insurance.

Why does p 7 line 16 state that info was collected about prior admissions for pneumonia when this group of patients was excluded according to the text and Figure 1?

I had difficulty linking a clinical relationship to pneumonia readmission rates and some of the factors on the list of clinical characteristics detailed on p7 line 20. Can the authors explain why they chose to include thrombocytosis and pressure ulcer as clinical covariates?

P9 line 5-8: More reason to exclude mention of the concept of late readmissions in this analysis. It adds little other than distracting from the main findings.

Figure 1 – graphic is blurred

Page 12 line 22, 23 – further explanation as to why inhaled bronchodilators were associate with increased risk of admission where as ICS-Bronchodilator combination was not

Figure 2 – no point estimate labeled for younger or older patients, therefor unable to visually compare point estimates between both groups

P13 line 1-6, the authors follow discussion of a finding that states that inhaled steroids were not a predictor of readmission with a reference to a study that found that inhaled steroids are associated with higher risk of developing pneumonia. Was this a study about pneumonia readmission rates or primary episodes of pneumonia? If the latter, then the link between the 2 thoughts does not logically connect. Consider rewriting or removing.

Since you have a separate paragraph addressing the relationship of systemic steroids and pneumonia readmissions (p13 line 17-18), consider removing mention of steroids in line 7 and just focus on ASA in this paragraph.

6. PLOS authors have the option to publish the peer review history of their article (what does this mean?). If published, this will include your full peer review and any attached files.

Reviewer #1: No

Reviewer #2: No

---

## [Author Response · Author response to Decision Letter 0]

20 May 2020

Reviewer #1: 

This study shows the factors associated with readmissions in a large cohort of patients with pneumonia. Also, this study adds evidence about complexity medication regimens as a possible predictive factor for readmission.

The follow-up of patients included early and late readmission (30 and 90 days). Unfortunately, the analysis of patients who had a late readmission were not presented. According to your study aimed, information should be provided regarding patient characteristics and associated factors with late readmissions.

Reply: We thank the reviewer for this suggestion. As the reviewer said, in our study the follow-up of patients included as outcome the first hospitalization for pneumonia; these rehospitalizations included both early (within 30 days of discharge after the first pneumonia admission) and late (between 31 and 90 days after discharge) readmissions for pneumonia. Therefore, the aim of our study was to evaluate the incidence and predictors of early and late readmissions due to pneumonia. 

However, from a first explorative analysis, as reported at page 9, lines 5-8 of the old version of our manuscript, it resulted that, from 31 to 90 days after the index date, only 47 patients (0.02%) experienced a late readmission for pneumonia. Thus, considering that the rate of late readmissions is very low, statistical analyses were carried out only for early readmissions. Statistical analyses were not carried out also for late readmissions for pneumonia because a lack of statistical power would have been observed, mainly due to the low number of events considered, which would not have allowed to highlight appreciable statistically significant differences in the predictor variables. 

Then, considering the potential lack of statistical power that could have occurred analyzing late pneumonia readmissions, and also taking into consideration the various suggestions raised by the reviewer #2 who proposed not to include late readmissions in the main analyses, we decided to remove from the main analyses the readmissions that occurred beyond 30 days after the index date. We revised the paper focusing on the early readmissions (i.e., those within 30 days after the index discharge for pneumonia), being these early readmissions the standard that most other research in this area examined.

However, we mentioned late readmissions a few times in the manuscript to broaden the discussion and introduce potential future developments of this research.

The study data were retrieved from the health care utilization database of NHS. Community-acquired pneumonia and hospital-acquired pneumonia were included? Please, add the pneumonia criteria and ICD-9CM codes included of index admission. Interestingly, perhaps you have included information about type of pneumonia and causative microorganism.

Reply: As the Reviewer suggested, our study is based on HealthCare Utilization (HCU) databases on Lombardy region, and all the 266,766 residents in Lombardy region assisted by the NHS wo had experienced at least one hospital admission with primary or secondary diagnosis of pneumonia, during the years 2003-2012, were firstly identified to be included in the study. Hospitalizations with pneumonia as secondary diagnosis were also considered because both the primary and secondary diagnoses contributed decisively in the hospital admission decision. As diagnosis of pneumonia for the index hospitalization, we considered those that had been recorded with an ICD-9-CM code from 480.x to 488.x, excluding 487.x but including 4870.x. 

We considered also hospital admissions with primary or secondary diagnosis with the ICD-9-CM code 486.x, so we could reasonably state that both Hospital-Acquired and Community-Acquired Pneumonia (HAP and CAP, respectively) has been included in the index hospitalization, since the ICD-9-CM code 486.x is used to record also community acquired pneumonia, healthcare associated pneumonia and hospital acquired pneumonia.

Furthermore, since the major outcome was the first rehospitalization for pneumonia within 30 days after the discharge from the first pneumonia admission, both HAP and CAP had been included also in this category.

As requested by the reviewer, we added in the manuscript the pneumonia ICD-9-CM codes that were considered for the identification of the index hospitalization [Materials and Methods section, Setting subsection, page 6, lines 19-20; Outcome subsection, page 7, lines 9-10].

Further information about type of pneumonia and causative microorganism were not available, since one of the major limitations of our study is that the information recorded in HCU databases did not include clinical data as causative microorganism, etc. 

You need explain if a collinearity analysis was performed. Why did you not show the use of statins (Table 1)?

Reply: We have considered collinearity in the context of a statistical model that is used to estimate the relationship between one response variable and a set of predictor variables; multicollinearity occurs when there are high correlations among predictor variables, leading to unreliable and unstable estimates of regression coefficients. We performed a collinearity analysis considering most widely-used diagnostic for multicollinearity, the Variance Inflation Factor (VIF). VIF has been calculated for each covariate by performing a linear regression model of that predictor on all the other covariates. The VIF has been calculated using the following formula: 1/(1-R2); with R2 estimated from the above-mentioned linear regression model. This parameter, i.e., VIF, estimates how much the variability of a coefficient is inflated because of linear dependence with other predictor variables. The VIF has a lower bound of 1 but no upper bound; in order to classify VIF as “High”, we fixed the threshold of 3. We included in the main analysis only those predictor variables that did not have high VIFs.

According to the reviewer’s suggestion, information regarding the previous use of lipid-lowering and antihypertensive medications has been added in Table 1 [pages 23-24]

Tables, figures, and references need a review (references duplicate nº 5 and 14)

Reply: According to the suggestion of the reviewer, we had carefully read the manuscript many times, particularly focusing on Tables, Figures and References. References duplicated has been corrected, and several modifications has been apported to tables, figures, figures legends and references’ format. Furthermore, Table 1 has been consistently modified [pages 23-24]. Also, Table 2 [pages 25] and Figure 2 were consistently revised.

The introduction and discussion need a little information regarding frail elderly patients.

Reply: We thank the reviewer for this advice. According to his suggestion, the topic of frail patients has been introduced and discussed thoughtfully in the manuscript, and it has been now reported in the Introduction and Discussion section [Introduction, Page 5, Lines 1-8; Discussion, Page 14, Lines 4-11; Page 16, lines 23-24; Page 17, lines 1-2]

Reviewer #2: 

The authors present a retrospective review of a large cohort of patients admitted with pneumonia to multiple hospitals in Northern Italy between 2003 and 2012 in which they assess the rate of pneumonia-specific readmission and its contributing cofactors. As opposed to most other studies on pneumonia readmission rates, the authors focus on related, pneumonia-specific readmissions, rather than all-cause readmissions. This is a strength of the study. Limitations include the significant delay between data collection and manuscript.

Major critiques

The authors should explain why they feel it is relevant to consider readmissions that occur beyond 30 days in their analysis. Intuitively it seems that these so called “late readmissions” are unrelated to factors involved in the original case. Furthermore, the rate of late readmissions is very low. Unless the authors can add further justification for including this in the paper, I would suggest removal and focusing on the readmissions within 30 days which is the standard most other research in this area examines.

Reply: We thank the reviewer for this suggestion. As the reviewer said, the aim of our study was to evaluate the incidence and predictors of early and late readmissions due to pneumonia. 

However, from a first explorative analysis, as reported at page 9, lines 5-8 of the old version of our manuscript, it resulted that, from 31 to 90 days after the index date, only 47 patients (0.02%) experienced a late readmission for pneumonia. Thus, considering that the rate of late readmissions is very low and the potential lack of statistical power that could have occurred analyzing separately late pneumonia readmissions, according to the reviewer suggestion we decided to remove from the main analyses the readmissions that occurred beyond 30 days after the index date. We revised the paper focusing on the early readmissions (i.e., those within 30 days after the index discharge for pneumonia), being these early readmissions the standard that most other research in this area examined.

However, we mentioned late readmissions a few times in the manuscript to broaden the discussion and introduce potential future developments of this research.

The data being analyzed in this study is currently 8 – 17 years old. Certainly, temporal changes in the care of hospitalized patients have transpired, and the fact that the conclusions are based on old data should be clarified as a limitation in the discussion sections. Ideally the authors would frame their results against comparative readmission rates from more recent data and put their results into context. Unfortunately, few studies exist with pneumonia specific readmission rates, but if the authors have access to all-cause 30-day readmission rates, they could make the case for whether temporal trends suggest increasing, decreasing or stable rates compared to the study period.

Reply: we thank the reviewer for this advice. We discussed this topic in the Discussion section regarding the limitations of the study [page 16, lines 16-21]. 

Furthermore, as suggested by the reviewer, all-cause 30-day readmission rate was calculated and widely discussed along the manuscript [page 10, lines 4-5; page 13, lines 5-10; page 16, lines 16-21].

As suggested by the reviewer the data analyzed in this study relate to several years ago and are not updated, for this reason our study may not have been able to include some changes that have recently occurred in the care of hospitalized patient for pneumonia. However, our all-case 30-day readmission rates after a first pneumonia hospitalization agree with those reported in recent observational studies, ranging from 8.6 to 25% [6,8,19-20], thus we can likely suppose that all-case 30-day readmission rates after a first pneumonia hospitalization remained stable

In Table 1, there appears to be significant differences in various comorbidities between readmitted and not readmitted patients; this is pointed out in line 15 on p 9. However, when adjusted hazard ratios are calculated (Table 2) only renal disease appears to be a predictor. In the Results section (p10, line 4-7) the authors note that use of meds for chronic lung disease were associated with increased readmission risk, but chronic lung disease as a comorbidity was not significant. Aren't some of these meds in fact "proxies" for the underlying chronic medical conditions? The authors should discuss these dichotomies and the implications for the limitations of their data in the discussion section.

Reply: We thank the reviewer for this suggestion. Reading the manuscript and checking the results, we found that a result was incorrectly reported: in Table 2 the Hazard ratios (HR), and relative 95% confidence intervals (CI), of early rehospitalizations for pneumonia for patients having Chronic lung disease was not 0.95 (0.88-1.02), but the correct estimates (now reported in Table 2) are 1.12 (1.03-1.25). The appropriate and related changes were therefore made in the text [Page 7, lines 24-25; Page 11, lines 5-6; Table 2; Figure 2].

The paper would benefit from more careful proofreading and correction of grammatical errors, and some effort to improve the clarity of writing.

Reply: We thank the reviewer for this advice. According to this suggestion, we had the proofs of the manuscript corrected by a professional translator Ms. Mary McKenney.

Data Table 1 lacks column headings, these should be added.

Reply: According to the reviewer’s suggestion, Table 1 has been consistently modified, and also column headings have been added.

Minor critiques

Late readmissions are mentioned in the introduction but given the lack of clear clinical relevance and the exceedingly small number identified by the author, I think the paper would be more effective if this detail was omitted throughout the manuscript.

Reply: We thank the reviewer for this suggestion. According to this, as better explained in the response to his first suggestion in the Major critiques section, we revised the paper focusing on the early readmissions (i.e., those within 30 days after the index discharge for pneumonia), being these the standard that most other respiratory medicine research examines. However, we mentioned late readmissions a few times in the manuscript to broaden the discussion and introduce potential future developments of this research.

The statement about readmission penalties in the US is inaccurate in that it refers to the “private health insurance system.” In fact, 30-day readmission penalties in the US are imposed by Medicare which is a federally funded health insurance program for adults age 65 and older. It is not private insurance.

Reply: We agree with the suggestion of the reviewer and the statement about readmission penalties in the US previously referring to the “private health insurance system” has been rewritten [page 5, lines 22-23; page 6, lines 1-2].

Why does p 7 line 16 state that info was collected about prior admissions for pneumonia when this group of patients was excluded according to the text and Figure 1?

Reply: We thank the reviewer for his suggestion, this step of the cohort selection procedure is not very clear and misleading. According to the reviewer suggestion the sentence “previous admissions for pneumonia (up to 6 months before the index date)” was deleted.

We still want to highlight that patients who experienced a hospitalization for pneumonia in the 6 months before the index date has been excluded; with this cautionary criterium we ensured the inclusion in the cohort of patients for whom the index admission was not a rehospitalization after prior recent pneumonia hospital admission, avoiding a potential misclassification of the outcome. 

I had difficulty linking a clinical relationship to pneumonia readmission rates and some of the factors on the list of clinical characteristics detailed on p7 line 20. Can the authors explain why they chose to include thrombocytosis and pressure ulcer as clinical covariates?

Reply: We thank the reviewer for this comment. We made a selection of covariates starting from the scientific literature (Weinreich M et al. Predicting the Risk of Readmission in Pneumonia: A Systematic Review of Model Performance. Ann Am Thorac Soc. 2016 Sep;13(9):1607-14; Makam AN et al. Predicting 30-Day Pneumonia Readmissions Using Electronic Health Record Data Journal of Hospital Medicine 2017 Vol 12. No 4). Furthermore, thrombocytosis is often used as a proxy of activated inflammatory state and pressure ulcers are more common in frail patients, a population analyzed in our study.

P9 line 5-8: More reason to exclude mention of the concept of late readmissions in this analysis. It adds little other than distracting from the main findings.

Reply: According to the reviewer this suggestion, and the previous ones, this sentence has been deleted.

Figure 1 – graphic is blurred.

Reply: According to the reviewer’s suggestion, Figure 1 has been modified and saved in a new format, in order to be clearer and sharper.

For a better view of the Figure 1, from the PDF file submitted to the Editorial Manager of the Journal, we please the reviewer not to view the Figure 1 directly from the PDF of the submission, but by going to the link (top right, “Click here to access / download ; Figure; Figure 1. Flow-chart.tiff”) and view the image after having downloaded it from the corresponding webpage; by doing this, the resolution of the image of Figure 1 will be higher.

Page 12 line 22, 23 – further explanation as to why inhaled bronchodilators were associate with increased risk of admission whereas ICS-Bronchodilator combination was not.

Reply: We thank the reviewer for this comment. This result may be due to the effect of the association with inhaled steroids. In fact, inhaled steroids were not associated with an increased risk of hospital readmission.

However, this is a population study based on HCU databases and not a pharmacological study, therefore the effect of individual molecules or associations cannot be assessed.

Figure 2 – No point estimate labeled for younger or older patients, therefore unable to visually compare point estimates between both groups

Reply: According to the reviewer suggestion, Figure 2 has been modified labelling the point estimates for younger and older patients.

P13 line 1-6, the authors follow discussion of a finding that states that inhaled steroids were not a predictor of readmission with a reference to a study that found that inhaled steroids are associated with higher risk of developing pneumonia. Was this a study about pneumonia readmission rates or primary episodes of pneumonia? If the latter, then the link between the 2 thoughts does not logically connect. Consider rewriting or removing.

Reply: We thank the reviewer for this observation and removed from the text the citation.

Since you have a separate paragraph addressing the relationship of systemic steroids and pneumonia readmissions (p13 line 17-18), consider removing mention of steroids in line 7 and just focus on ASA in this paragraph.

Reply: We agree with the reviewer on this suggestion and changed the text accordingly.

---

## [Decision Letter · Decision Letter 1]

9 Jun 2020

PONE-D-19-32710R1

Rehospitalization for pneumonia after first pneumonia admission: Incidence and predictors in a population-based cohort study

PLOS ONE

Dear Dr. Faverio,

Thank you for submitting your manuscript to PLOS ONE. After careful consideration, we feel that it has merit but does not fully meet PLOS ONE’s publication criteria as it currently stands. Therefore, we invite you to submit a revised version of the manuscript that addresses the points raised during the review process

Thank you for taking the time to revise and resubmit your manuscript to PLOS ONE. As you will see, both of the Reviewers were satisfied with the revisions and only one has pointed out that the Abstract needs to be revised to reflect the updated Results. In addition to revisions to the Abstract, I would also like to see the following adjustments:

- In your Discussion, please remove any reference to the late pneumonia readmissions. This is never really addressed earlier in the manuscript (at least not now with the changes) so I think it should now be removed from here as well. This modification will also require some changes to the paragraph starting line 20 on page 14.

We look forward to receiving your revised manuscript.

Kind regards,

Andrea Gruneir

Academic Editor

PLOS ONE

Reviewers' comments:

Reviewer's Responses to Questions

**Comments to the Author**

1. If the authors have adequately addressed your comments raised in a previous round of review and you feel that this manuscript is now acceptable for publication, you may indicate that here to bypass the “Comments to the Author” section, enter your conflict of interest statement in the “Confidential to Editor” section, and submit your "Accept" recommendation.

Reviewer #1: All comments have been addressed

Reviewer #2: All comments have been addressed

2. Is the manuscript technically sound, and do the data support the conclusions?

Reviewer #1: Yes

Reviewer #2: Yes

3. Has the statistical analysis been performed appropriately and rigorously? 

Reviewer #1: Yes

Reviewer #2: I Don't Know

4. Have the authors made all data underlying the findings in their manuscript fully available?

Reviewer #1: Yes

Reviewer #2: No

5. Is the manuscript presented in an intelligible fashion and written in standard English?

Reviewer #1: Yes

Reviewer #2: Yes

6. Review Comments to the Author

Reviewer #1: This paper has improved substantially focusing the analysis on patients readmitted within 30 days.

The authors have adequately addressed the reviewer comments.

Minor critiques:

The authors should review abstract, objective does not correspond with the main text (predictors of late readmissions were not evaluated).

Reviewer #2: My comments were addressed sufficiently and the paper now appears to be appropriate for publication in my opinion.

7. PLOS authors have the option to publish the peer review history of their article (what does this mean?). If published, this will include your full peer review and any attached files.

Reviewer #1: No

Reviewer #2: No

---

## [Author Response · Author response to Decision Letter 1]

10 Jun 2020

Academic Editor: In your Discussion, please remove any reference to the late pneumonia readmissions. This is never really addressed earlier in the manuscript (at least not now with the changes) so I think it should now be removed from here as well. This modification will also require some changes to the paragraph starting line 20 on page 14.

We thank the Academic Editor for this suggestion. We changed the text accordingly.

Reviewer #1: This paper has improved substantially focusing the analysis on patients readmitted within 30 days.

The authors have adequately addressed the reviewer comments.

Minor critiques:

The authors should review abstract, objective does not correspond with the main text (predictors of late readmissions were not evaluated).

We thank the reviewer for this suggetion. We changed the text accordingly.

Reviewer #2: My comments were addressed sufficiently and the paper now appears to be appropriate for publication in my opinion.

Thank you

---

## [Editor Report · Decision Letter 2]

17 Jun 2020

Rehospitalization for pneumonia after first pneumonia admission: Incidence and predictors in a population-based cohort study

PONE-D-19-32710R2

Dear Dr. Faverio,

We’re pleased to inform you that your manuscript has been judged scientifically suitable for publication and will be formally accepted for publication once it meets all outstanding technical requirements.

Kind regards,

Andrea Gruneir

Academic Editor

PLOS ONE
---

## [Editor Report · Acceptance letter]

19 Jun 2020

PONE-D-19-32710R2 

Rehospitalization for pneumonia after first pneumonia admission: Incidence and predictors in a population-based cohort study 

Dear Dr. Faverio:

I'm pleased to inform you that your manuscript has been deemed suitable for publication in PLOS ONE. Congratulations! Your manuscript is now with our production department. 

Kind regards, 

on behalf of

Dr. Andrea Gruneir 

Academic Editor

PLOS ONE